# Structure of *Vibrio* Phage XM1, a Simple Contractile DNA Injection Machine

**DOI:** 10.3390/v15081673

**Published:** 2023-07-31

**Authors:** Zhiqing Wang, Andrei Fokine, Xinwu Guo, Wen Jiang, Michael G. Rossmann, Richard J. Kuhn, Zhu-Hua Luo, Thomas Klose

**Affiliations:** 1Department of Biological Sciences, Purdue University, West Lafayette, IN 47907, USA; 2National Cryo-EM Facility, Frederick National Laboratory for Cancer Research, Leidos Biomedical Research Inc., Frederick, MD 21701, USA; 3Sansure Biotech Inc., Changsha 410205, China; 4Key Laboratory of Marine Biogenetic Resources, Third Institute of Oceanography, Ministry of Natural Resources, Xiamen 361005, China; 5School of Marine Sciences, Nanjing University of Information Science and Technology, Nanjing 210044, China

**Keywords:** virus assembly, contractile injection system, *Myoviridae* tail, phage infection mechanism, baseplate, contractile sheath

## Abstract

Antibiotic resistance poses a growing risk to public health, requiring new tools to combat pathogenic bacteria. Contractile injection systems, including bacteriophage tails, pyocins, and bacterial type VI secretion systems, can efficiently penetrate cell envelopes and become potential antibacterial agents. Bacteriophage XM1 is a dsDNA virus belonging to the *Myoviridae* family and infecting Vibrio bacteria. The XM1 virion, made of 18 different proteins, consists of an icosahedral head and a contractile tail, terminated with a baseplate. Here, we report cryo-EM reconstructions of all components of the XM1 virion and describe the atomic structures of 14 XM1 proteins. The XM1 baseplate is composed of a central hub surrounded by six wedge modules to which twelve spikes are attached. The XM1 tail contains a fewer number of smaller proteins compared to other reported phage baseplates, depicting the minimum requirements for building an effective cell-envelope-penetrating machine. We describe the tail sheath structure in the pre-infection and post-infection states and its conformational changes during infection. In addition, we report, for the first time, the in situ structure of the phage neck region to near-atomic resolution. Based on these structures, we propose mechanisms of virus assembly and infection.

## 1. Introduction

*Vibrio* is a genus of Gram-negative bacteria that have curved-rod shapes and a polar flagella with a sheath [1]. *Vibrio* species are typically found in salt water. Pathogenic strains of *Vibrio* bacteria cause gastroenteritis in humans through contaminated water or under-cooked seafood [2]. Among these, *V. cholerae* is the causative agent of the deadly cholera disease. Other strains, such as *V. harveyi*, can cause diseases in aquatic animals [3]. The native control agents for *Vibrio* bacteria are bacteriophages (or phages). Many *Vibrio* phages have been isolated and characterized by genome sequencing in recent decades [4,5,6]. Among these, only a few have been studied using structural biology methods, such as Siphophage SIO-2 [7].

Bacteriophage XM1 was isolated from an abalone farm in Xiamen, China. Its hosts were identified as *V. harveyi* and *V. rotiferianus*. XM1 is a double-stranded DNA virus that belongs to the *Myoviridae* family of the Caudovirales order. The XM1 virion has an icosahedral capsid, or head (Figure 1a), with a diameter of ~640Å, which encapsulates a 38 kbp double-stranded DNA genome. The capsid has one special vertex occupied by a dodecameric portal protein through which the DNA enters during virus assembly and exits during infection. The XM1 phage possesses a ~900Å-long contractile tail attached to the special portal vertex of the capsid (Figure 1a). The tail is a molecular machine designed to penetrate the bacterial cell envelope and create a channel through which the genome is injected into the host. The XM1 tail consists of the baseplate, tail tube, and contractile sheath wrapping the tube.

The most well-characterized member of the *Myoviridae* family is bacteriophage T4, whose structure has been extensively studied for decades [8,9,10,11,12]. The contractile tail of phage T4 has structural similarity to the bacterial type VI secretion systems (T6SS) [13], and the bacterial tailocins [14,15]. T6SSs are found in Gram-negative bacteria and are used to transfer material from the interior of a bacterial cell across the cell envelope into an adjacent target cell [16]. The bacterial tailocins are secreted bacteriocins acting as puncturing devices that penetrate cell envelopes to dissipate the membrane potential and kill the attacked cell [17]. Altogether, the *Myoviridae* phage tails, T6SSs, and bacterial tailocins are referred to as the contractile injection systems (CISs) [18]. Cryo-electron microscopy (cryo-EM) reconstructions of two other *Myoviridae* phages, Φ812 [19] and A511 [20], were reported at intermediate resolutions. Even though their hosts are Gram-positive bacteria, their tails share a common structural organization with that of T4.

Bacteriophage XM1 has one of the most simple and compact contractile tails analyzed thus far. The high-resolution cryo-EM study provided here improves our understanding of the minimal structural components required to build an effective cell-envelope-penetrating machine. Here, we report a 3.2Å-resolution 6-fold-symmetric reconstruction of the distal region of the XM1 tail. This includes the baseplate and part of the tail tube, surrounded by a contractile sheath. Using cryo-EM, we mapped atomic structures of the major components of the baseplate, tail tube, and tail sheath. This allowed us to demonstrate that the six wedges of the XM1 baseplate interact with its central hub in a new manner not previously observed in other CISs. We also report here 3.6Å-resolution 12-fold and 6-fold-symmetric reconstructions of the phage portal and neck region, where the tail joins the head. Based on these cryo-EM reconstructions, we built the in situ atomic structure of the entire XM1 portal and neck complex which, to our knowledge, is the first such structure shown for a *Myoviridae* phage. Also, we produced a 9.1Å-resolution reconstruction of the tail and neck region of empty XM1 particles with the tail sheath contracted. This reconstruction represents the tail sheath in its post-infection form and reveals conformational changes that occur during infection. In addition, we determined the 3.2Å-resolution structure of the icosahedral XM1 capsid and further built the atomic structures of the major capsid protein and the decoration protein attached to the capsid. Based on these XM1 structures, we also describe the mechanisms of virus assembly, stability, and infection.

## 2. Results and Discussion

Organization of the XM1 baseplate. The XM1 baseplate is located at the distal end of the phage tail (Figure 1c) and consists of 78 polypeptide chains encoded by eight different genes. The cryo-EM structure of the phage XM1 baseplate was resolved to 3.2Å resolution using 6-fold symmetry averaging. Based on the cryo-EM maps, the atomic structures of gene products (gp)11, gp12, gp15, gp16, and gp17, constituting the baseplate, were manually built and refined. The locations and functions of the three other baseplate proteins, gp13, gp14, and gp18, were identified based on their homology to proteins from other phages and CISs, detected using HHpred sequence analysis [21] (Appendix A). Similar to T4 and other phage baseplates [12,22], the XM1 baseplate can be subdivided into the central hub, six baseplate wedges, and peripheral attachments.

The XM1 central hub contains three stacks of proteins located underneath the tail tube protein gp7. The stack right below the tail tube is formed by a hexameric ring of gp11, whose atomic structure is well resolved in the cryo-EM density. However, the structures of the two other stacks below the gp11 ring were not resolved because they did not follow the 6-fold symmetry imposed during the reconstruction process. In other phages and CIS structures, these stacks correspond to the trimeric hub protein and the trimeric cell-piercing protein. The XM1 gp13 protein has sequence homology to the trimeric hub protein of the R2 pyocin (PDB: 6U5H_A), suggesting that it is the central baseplate hub protein located underneath the gp11 ring [23]. The XM1 gp14 protein is homologous to the phage SN cell-piercing protein (PDB: 4RU3_A) suggesting that gp14 forms the most distal stack of the XM1 baseplate (below gp13) and functions as the cell-piercing device during infection.

Each of the six baseplate wedges is made of one copy of gp17, two copies of gp16, one copy of gp15, and one copy of gp12 (Figure 2a). The (gp16)_2_–gp17 complex is homologous to the (gp6)_2_–gp7 baseplate wedge module of phage T4 [22], whereas XM1 gp15 is homologous to T4 gp25, which serves as the initiator for assembly of the contractile tail sheath. The globular gp12 protein has an immunoglobulin-like fold and is harbored in the baseplate between the gp11 ring and gp16 dimers (Figure 2b).

Unlike phage T4, XM1 does not have a complicated tail fiber system attached to its baseplate. However, the cryo-EM map shows twelve oval-shaped densities attached to the XM1 baseplate periphery, with two such densities attached to each of the six baseplate wedges (Figure 2c). The sequence of the XM1 gp18 protein is homologous to the trimeric tail spike protein of phage vb_AbaP_AS12 (PDB: 6EU4_B), suggesting that gp18 forms the XM1 tail spike protein (TSP) corresponding to the densities observed in the cryo-EM map.

The central baseplate organization protein, gp11. The gp11 protein, located immediately underneath the tail tube, consists of 250 amino acids. Six copies of gp11 form a hexameric ring that continues the tail tube. The inner part of the ring has structural homology to the XM1 tail tube protein, gp7, and to the tail tube proteins of other CISs. The gp11 protein has a β-sandwich fold as its main structural framework and contains extended loops to interact with surrounding proteins (Figure 2a). Each gp11 subunit holds two adjacent gp15 monomers, one each to the left and right, and interacts with gp7 and gp13 on its top and bottom, respectively. A loop-like insertion, residues 85–127, interacts with the immunoglobulin-like gp12 protein and with the (gp16)_2_-gp17 modules. The gp11 C-terminus (residues 239–250) extends and wraps around the gp12 globule. The average interface area between gp11 and gp12 is ~1906Å^2^ (Table 1). In total, gp11 interacts with seven other proteins (gp6, gp7, gp12, gp13, gp15, gp16, and gp17; Table 1) and, therefore, plays a central role in baseplate organization.

The baseplate wedge scaffolding complex, (gp16)_2_–gp17. The two major baseplate components, gp16 and gp17, form the baseplate wedge scaffold. This part of the baseplate structure is well conserved in T4 [12,22], T6SS, and other CISs. Two symmetrically independent copies of gp16 attach to one copy of gp17 in each wedge, with six wedges forming the outer layer of the baseplate.

The gp16 molecule (Appendix A), consisting of 404 residues, can be divided into three domains. Domain I of gp16 (DI, residues 1–86 and 179–196) contains an N-terminal helix–turn–helix motif formed by residues 1–68. This motif interacts with the same motif as that of a neighboring gp16 subunit and with a similar N-terminal helix–turn–helix motif of gp17 (residues 1–48), thus forming the (gp16)_2_-gp17 “core bundle”. Two gp16 helices, residues 69–86 and 179–192, sit at the bottom of the core bundle and are involved in formation of the trifurcation region (Figure 2a). Domain II of gp16 (DII, residues 87–178) is inserted into DI and serves as a wing-like domain that covers the gp12 protein inside the baseplate, while domain III (DIII, residues 192–404) is homologous to region 340–489 of T4 gp6. The loop connecting DI to DIII (residues 192–198) extends differently in the two symmetrically independent copies of gp16, making each adopt a different conformation. DIII gp16 dimerizes with DIII gp16 from a neighboring baseplate wedge, thus contributing to the overall baseplate stability.

The XM1 gp17 protein consists of only 242 residues, whereas its T4 ortholog (gp7) consists of 1032 residues. XM1 gp17 contains flexible parts, which are involved in tail spike binding but were not resolved in the cryo-EM map. Therefore, only the atomic structures of gp17 residues 1–79 and 107–200 were built. DI of gp17 (residues 1–79 and 107–133) is homologous to DI of gp16, which has a helix–turn–helix motif participating in the core bundle formation. The two missing regions (residues 80–106 and 201–242) of the gp17 protein are embedded into the tail spike density and are responsible for interaction with the TSP, gp18. There are 12 gp18 spikes attached to the baseplate via gp17. One set of six spikes, oriented ~75° upward to the baseplate plane, is attached to the loops of residues 80–106 of the gp17 subunits, and the other set of six spikes, roughly in-plane with the baseplate, is attached to the C-terminal regions of gp17.

Immunoglobulin-like protein, gp12. The immunoglobulin-like molecule gp12 (188 residues, Figure 2a), enclosed inside the baseplate, holds tightly to the gp11 protein and interacts with DI of gp16. Unlike other XM1 baseplate proteins, individual gp12 subunits do not interact with other gp12 subunits in the baseplate. The gp12 protein is a unique feature of the XM1 baseplate, as no orthologs of this protein have been found in other known baseplate structures in terms of the atomic structure, location, and interactions with surrounding proteins.

Tail sheath polymerization initiator, gp15. The monomeric protein gp15 (117 residues) lies on top of each (gp16)_2_–gp17 core bundle and is homologous to the phage T4 tail sheath assembly initiator protein (gp25). The atomic structures of the gp15 subunits were built and refined, with no corresponding density to the seven C-terminal residues. The overall structure of gp15 resembles the C-terminal domain (CTD) of the XM1 contractile sheath protein gp6 (Figure 3d and Figure 4c). The gp15 forms a binding platform that holds the N- and C- termini of the two neighboring sheath subunits from the first sheath ring. The gp15-sheath interaction interface has an area of ~1642Å^2^ (Table 1). In the baseplate, gp15 functions as the initiator of polymerization of the sheath protein and as the binding platform to stabilize the first ring of the sheath. The gp15 protein also interacts with the two adjacent gp11 subunits with an interface area of ~1711Å^2^ (Table 1). Meanwhile, gp15 attaches to the (gp16)_2_–gp17 module with an interface area of ~1615Å^2^ (Table 1) in total. Located at the junction of the baseplate wedge and tail sheath protein, gp15 protein may transduce the signal from the baseplate periphery to the sheath and trigger sheath contraction during infection.

Tail tube and tail sheath structures. The tail tube protein, gp7, forms a stack of 18 hexameric rings on top of the baseplate. Each ring has an inner diameter of ~40Å and an outer diameter of ~80Å. The tail tube protein has a β-sandwich fold similar to other CIS tail tube proteins, such as gp19 of phage T4.

The XM1 tail sheath protein, gp6, also forms a stack of 18 hexameric rings surrounding the tail tube. In the pre-infection (extended) state of the tail, each sheath protein ring has inner and outer diameters of ~80Å and ~205Å, respectively. In the extended XM1 tail, the tube and sheath rings are arranged into a helix, with each subsequent ring in the stack being rotated by 27° and translated by 35Å relative to the previous ring.

The tail sheath protein, gp6, contains 495 amino acid residues. The quality of the cryo-EM map allowed for the resolution of the atomic structure of the entire sheath protein structure, which consists of three domains (Figure 3d). The CTD contains antiparallel β-strands and two long α-helices. This CTD is well conserved in contractile systems as has been observed in the *Photorhabdus* virulence cassette (PVC), antifeeding prophage (AFP), and T6SS sheath proteins VipA/B [13,14,15,25]. The N-terminal domain (NTD) has an αβα-sandwich fold in which the middle six β-strands are covered by two layers of three α-helices on each side. The structure of NTD is homologous to that of the PVC and AFP sheath proteins. However, in T6SS VipB, this domain has a similar αβα-sandwich fold with a different topology, indicating that the XM1 phage sheath protein is evolutionarily closer to those of PVC and AFP [13,14,15]. The sheath protein of XM1 has an insertion domain (InD) that is inserted into the NTD. The InD is an oval-shaped structure unique to phage XM1. A Dali structural homology search [26] detected a similarity between the InD and D3 domain of flagellar cap protein (FliD; PDB:5H5V). Since a flagellar cap is a common feature of the host *Vibrio* bacteria, this domain might be obtained by the XM1 phage via gene exchange from its host during evolution.

In the extended tail, the sheath protein subunit is in an L-like “standing” orientation, with CTD pointing upward and the NTD and InD oriented horizontally (Figure 3d). The sheath protein N-terminus forms a parallel β-sheet with the C-terminus of a neighboring sheath subunit belonging to the same hexameric ring, forming a “handshake” interaction (Figure 3a–c). A similar “handshake” arrangement has been observed in the R-type pyocin structure and T6SS sheath [25,27]. Furthermore, this “handshake” β-sheet forms an augmented 4-stranded β-sheet with the CTD of the sheath subunit located in the lower hexameric ring, tightly holding the stacking rings of the sheath protein together. Thus, the CTD of the sheath subunit located in the lower ring provides a binding platform for the attachment of two neighboring sheath proteins in the upper ring.

About 5% of particles in our purified phage sample were empty. The 6-fold-symmetric reconstruction of the XM1 tail obtained from these empty particles reached a 9.1Å resolution. In these particles, the tail sheath was contracted (Figure 3e,f) and the baseplate was detached from the sheath (Figure 4a). The pre-infection (extended) structures of sheath protein subunits were fitted into the contracted sheath map and refined using the ddforge program from the Situs software package [28]. The extended and contracted sheath subunits can be superimposed with a root mean square deviation (RMSD) of ~1.3Å between 220 equivalenced Cα atoms (Figure 4b), indicating that during contraction, the sheath subunits mainly moved as rigid bodies. During contraction, each sheath protein subunit rotates from a “standing” to a “lying” orientation, with the CTD pointing inside and the InD pointing slightly downward (Figure 3h). Altogether, the hexameric rings of the sheath move vertically up and expand horizontally outward. Upon contraction, the length of the sheath shrinks from ~630Å to ~290Å. The sheath subunits detach from the tail tube and the inner and outer diameters of the sheath rings expand to ~100 and ~265Å, respectively. In the contracted sheath, each subsequent ring in the stack is rotated by 32° and translated by 16Å relative to the previous ring.

In the cryo-EM map of the contracted tail, the densities corresponding to the N- and C-terminal regions were faint, suggesting that these regions likely moved more during contraction. However, the density corresponding to the augmented β-sheet of the CTD binding platform was obvious (Figure 4e). Therefore, this augmented β-sheet is important in keeping the integrity of the tail sheath during contraction. A previous study of the R2-type pyocin in the pre- and post-infection states demonstrated a similar mechanism of sheath stabilization in the post-infection state [27].

The cryo-EM map of the contracted tail shows a density corresponding to the tail sheath initiator protein, gp15, underneath the gp6 sheath. Most of the gp15 structure, except the N-terminal region (residues 1–22), fits well into this density as a rigid body (Figure 4f). This result shows that the β-sheet augmentation in gp15 and gp6 is preserved before and after sheath contraction.

Organization of the portal and neck region. The cryo-EM reconstruction of the portal and neck region of the XM1 virion was calculated to 3.6Å resolution using 6-fold symmetry. The upper part of the reconstruction, corresponding to the portal, was further refined using 12-fold symmetry and resulted in a similar resolution. The XM1 portal and neck region contains 54 polypeptide chains, encoded by 5 different genes (Figure 1b and Figure 4a, Appendix A). Based on cryo-EM maps, the atomic structures of 12 gp49 subunits, 12 gp1 subunits, 6 gp4 subunits, 6 gp5 subunits, and partial structures of 18 gp40 subunits were manually built and refined.

Portal protein, gp49. The dodecamer of portal protein gp49 has a flying saucer shape with a central channel through which DNA enters the capsid during virus assembly and exits during infection. Each gp49 subunit consists of 412 amino acids, from which we were able to build and refine residue regions 8–337 and 346–378. The XM1 gp49 structure is homologous to the portal proteins of phages g20c (PDB:4ZJN) and T4 (PDB:3JA7) (Figure 5b). The structure of the portal protein subunits consists of typical clip, stem, wing, and crown domains [29]. The cryo-EM map does not show densities corresponding to the seven N-terminal residues of the portal subunits. This might indicate that the N-terminal regions do not obey the applied 12-fold symmetry and that their conformations are different in each portal subunit. In phage T4, the N-terminal regions of the portal protein subunits morph to compensate for the symmetry mismatch between the 5-fold-symmetric capsid and the 12-fold-symmetric portal. Therefore, the N-terminal regions cannot be resolved when 12-fold symmetry is applied [30]. Residue region 8–32 of the XM1 portal extends and wraps around four neighboring portal monomers to its left, suggesting that this region plays an important role in holding the portal dodecamer together. Positively charged amino acids, such as K352, K300, R273, and K266, lie on the surface of the inner channel of the portal, with their distance to the central DNA density being less than 10Å. These residues form several positively charged rings and are likely to prevent DNA backsliding.

Dodecameric head completion protein, gp1, attaches to the portal. The dodecameric ring underneath the portal protein is formed by gp1 (118 residues). The secondary structure of gp1 is dominated by α-helices (Figure 5c). The gp1 protein attaches to the gp49 portal by its C-terminal region (residues 113–117), inserted between two α-helices from two stem domains of neighboring portal subunits. The fold of gp1 is homologous to that of other phage connector proteins, such as gp36 of phage Mu (PDB: 5YDN), gp15 of SPP1 (PDB:5A21_C), and gp6 of HK97 (PDB:3JVO). Compared to Mu gp36 and HK97 gp6 [31,32], the XM1 gp1 protein has a long insertion region, residues 62–100, formed by a pair of antiparallel β-strands and a short helix. These antiparallel β-strands make the inner surface of the gp1 ring and face the DNA double helix (Figure 5c). Five serine residues (73, 75, 78, 80, 82), located on the β-turn-β structure, are facing inside. The shortest distance between the two serine residues on the opposite side of the gp1 ring is ~30Å. Considering the diameter of the DNA double helix is ~20Å, the estimated distance from one of the serines to the closest DNA phosphate group is ~5Å. These serine residues might be involved in hydrogen bonding with the phosphate groups of the DNA double helix. 

Neighboring serine residues along the same β-strand is ~7Å, which roughly corresponds to two pitches of the DNA helix. This unique structure arrangement suggests a new way of gating the DNA passage channel.

Hexameric head completion protein, gp4, forms the tail attachment interface. Six gp4 subunits (114 residues) form a hexameric ring just below gp1. The gp4 protein has a β-sandwich fold and shows structural homology to the phage SPP1 gp16 protein (PDB: 2KCA), located in a similar position in the phage neck. In SPP1 gp16, the 19-residue-long loop between strands β2 and β3 protrudes inward, facing the DNA double helix [33]. Therefore, it was proposed that SPP1 gp16 seals the portal vertex and stops the DNA from leaking. Phage XM1 gp4 also has a 15-residue-long (residues 46–61) loop located between β2 and β3, however, it does not show sequence homology to the loop of SPP1 gp16. There are three lysine residues (48, 49, 50) lining the inside of the gp4 loop (Figure 5d). The distance between two lysine 48 side chains on the opposite side of the gp4 hexamer ring is ~28Å. This suggests that the positively charged ring made by the lysine residues may constrict DNA movement.

Tail terminator protein, gp5. The XM1 gp5 protein (161 residues) forms a hexameric layer underneath the gp4 ring and above the tail tube protein. This gp5 structure is homologous to the tail terminator proteins of phages lambda (gpU, PDB: 3FZ2_E), T4 (gp15, PDB: 3J2M), and SPP1 (gp17, PDB:5A21_G). Among these phages, only T4 has a contractile sheath. The T4 gp15 structure, solved by X-ray crystallography, did not contain the C-terminal region of the protein [34]. In phage XM1, the C-terminus of gp5 extends into the density of tail sheath protein gp6 and forms a β-sheet with the β-sheet of the sheath protein (Figure 5e). This interaction is analogous to the β-sheet augmentation observed in the tail sheath. The interface area between gp5 and sheath protein in the extended tail is ~1387Å [2] (Table 1).

The gp5 hexamer and the modeled contractile gp6 sheath hexamer were fitted into the cryo-EM map of the contracted tail (Figure 4d). Even though the map is at 9Å-resolution, the fitting indicated that the gp5 hexamer remained in the same position as it was in the pre-infection state, while gp6 moved and rotated into its post-infection state. The relative position change indicates that the gp5 C-terminal region might have moved together with the sheath CTD as part of the β-sheet augmentation. This is supported by the fact that the sheath remained attached to the neck through the gp5 hexamer ring. However, the resolution of the map limited our efforts to trace the gp5 C-terminal region.

Collar spike protein, gp40. Six trimers of the gp40 protein (839 residues) are attached to the XM1 neck. Although most of the gp40 structure is flexible, it was possible to build atomic models corresponding to the N-terminal regions of gp40 subunits (residues 1–126). The gp40 subunits contain a 100-residue-long NTD, which is responsible for binding to the neck. This domain has an immunoglobulin-like (Ig-like) fold and is homologous to the collagen adhesion protein (PDB: 6FWV), with an RMSD of 2.6Å among 81 equivalent Cα atoms. The three NTDs belonging to different subunits of the same gp40 trimer show different modes of interaction with the neck proteins. In subunits ‘a’ and ‘b’ of gp40, the NTD is lined up with gp4 (Figure 5f). The interface area between gp4 and the gp40_a and gp40_b NTD together is ~950Å^2^ (Table 1). The NTD of subunit ‘c’ is rotated by about 120° relative to subunits ‘a’ and ‘b’ and forms interactions with the two neighboring gp5 subunits (Figure 5f), with an interface area of ~733Å^2^ (Table 1). The C-terminal portion (about two thirds) of the gp40 sequence is homologous to phage Det7 TSP, gp208 (PDB: 6F7D_A). The C-terminal tail spike-like region of gp40 is connected to its NTD via a 100-residue-long α-helix. Three such helixes from each gp40 subunit in the trimer form a coiled-coil structure. Because of its flexibility, only some of this coiled-coil structure (residues 101–126) was determined. Remarkably, the trimeric lower tail fiber protein (FibL) of phage 80α [35] also has an Ig-like NTD, which attaches to the virion, followed by a long α-helix forming a trimeric coiled-coil. The interactions of the three NTDs of the gp40 trimers with the gp4 head completion protein and the gp5 tail terminator protein reinforce the attachment of the tail to the capsid (Appendix A). The homology of gp40 to the TSPs suggests that gp40 may be involved in sensing hosts and/or the attachment of XM1 virions to the cell surface.

Structure of the XM1 capsid. The icosahedral XM1 head has a diameter of about 640Å. Cryo-EM reconstruction of the head using icosahedral symmetry reached a 3.2Å resolution (Figure 6a). The XM1 capsid shell is made of two proteins: the major capsid protein (MCP) and the minor capsid protein (or decoration protein, Dec). The XM1 MCP, encoded by gene 54, has a length of 324 residues, whereas the Dec protein, encoded by gene 53, consists of 160 residues. The N-terminal 30 residues of MCP were not detected in the cryo-EM map, suggesting that the MCP N-terminal region was cleaved off during capsid assembly, similar to many other tailed dsDNA phages [36,37].

MCP and Dec proteins are arranged into a hexagonal lattice characterized by the triangulation number T = 7 *laevo* (Figure 6b). Therefore, each asymmetric unit of the capsid contains seven MCP subunits and seven Dec subunits. The arrangement of the XM1 capsid proteins is similar to that in the marine phage TW1 [38].

The atomic structure of the asymmetric unit of the XM1 capsid was built and refined. The XM1 MCP protein has the typical HK97 fold with axial (A) and periphery domains, an N-terminal arm, and an extended loop [39]. The MCP subunits of XM1 and HK97 can be superimposed, with an RMSD of 1.3Å among 67 equivalent C-alpha atoms (Figure 6c). In phage XM1, a bowling pin-shaped density is present on the symmetry axes of the MCP hexamers and pentamers, where the tips of A domains are gathered together (Figure 6e). In phage HK97 capsomers, an interactive anion (SO_4_^2−^ or Cl^−^) is located in an equivalent position. However, a similar density at an equivalent position has not been reported in other phages with the HK97 fold. In phage XM1, there are two amino acids near this density, Asn235 and Gly236. When an Asn residue is followed by a Gly residue in the polypeptide chain, the Asn residue may form an L-succinimide with Gly. As a result, the Asn residue loses its -NH_2_ group and becomes an L-iso-aspartic acid [40]. This auto-deamination reaction is a non-enzymatic reaction that happens under certain physiological conditions (pH 7.4, 37 °C) [38]. The presence of L-iso-aspartic acid makes the protein susceptible to damage and is related to aging and neurological diseases [41]. It is interesting to not only observe an Asn-Gly pair in the phage MCP but also five or six such pairs clustering together near the centers of hexameric or pentameric capsomers. It is reasonable to propose that a cluster of NH_4_^+^ ions may reside at the capsomers’ centers or another small compound that inhibits auto-deamination reactions. Alternatively, other positively charged ions may reside in these positions because the surrounding iso-aspartic residues are negatively charged.

The minor decorative capsid protein, gp53, has a fold similar to the minor capsid protein of phage TW1 (PDB: 5WK1_K). These two proteins can be superimposed with an RMSD of 3.0Å among 126 equivalent Cα atoms (Figure 6d). Also, XM1 gp53 and TW1 Dec are arranged at almost identical positions relative to the major capsid proteins. Both TW1 minor capsid protein and XM1 gp53 form trimers on the capsid surface. The axes of the trimers coincide with the icosahedral 3-fold and quasi-3-fold axes, which relate neighboring capsomers. The NTD of gp53 (residues 1–95) predominantly interacts with the major capsid protein, whereas the CTD (residues 95–160) is mainly involved in gp53 trimerization. Like the phage TW1 minor capsid protein and phage lambda gpD [42], gp53 reinforces the capsid near the interfaces of three adjacent capsomers.

Assembly of the XM1 virion. Assembly of the tails and heads of *Myoviridae* phages proceeds via independent pathways and are joined together to form infectious virions [43]. Bacteriophage T4 is the most thoroughly investigated member of the *Myoviridae* family, whose assembly pathway is well-established [11,44]. Based on the structural similarity between the XM1 and T4 virions, we propose the following assembly pathway for phage XM1, which is analogous to that of T4.

The T4 tail assembly starts with the formation of the central region of the baseplate [11,22]. In phage XM1, the central baseplate region (Figure 2c) is formed by the trimeric proteins gp13 and gp14, the hexameric ring of gp11, as well as the tape measure protein, gp10. The central region of the baseplate serves as a hub to which six wedges are attached. XM1 structure shows that the central baseplate protein gp11 plays a key role in maintaining baseplate integrity by interacting with other protein components (Table 1). Even though there is no counterpart of gp12 found in phage T4, based on its position and interaction with gp11, the gp12 protein is likely the first recruited by gp11. Then the (gp16)_2_–gp17 modules attach to gp11 and gp12 to form the baseplate wedges (Figure 2c). These six wedges are then fastened together by the dimerization of gp16 DIII between two adjacent wedges. The gp11 hexamer, being homologous to the tail tube protein, also serves as a starting point for tail tube polymerization. Further, gp11, together with a (gp16)_2_–gp17 module, interacts with the gp15 sheath polymerization initiator protein. Altogether, this suggests that gp11 plays a key role in XM1 tail assembly (Figure 2c).

By analogy to T4 and other well-studied phages, the XM1 tail tube protein, gp7, polymerizes around the tape measure protein, gp10, making a stack of hexameric rings on top of gp11. Tail tube polymerization continues until the end of the tape measure protein is reached. The size of the tape measure protein, gp10 (479 residues), is roughly consistent with the tail tube length if the tape measure protein is modeled as a continuous α-helix. Cryo-EM reconstruction of the XM1 tail shows density in the center of the tail tube, which continues throughout the length of the tail tube and most probably corresponds to the gp10 protein.

The contractile sheath polymerizes around the tail tube. The sheath protein, gp6, makes a stack of hexameric rings, surrounding the tube, on top of the sheath initiator protein, gp15. Polymerization of the sheath continues until the end of the tail tube is reached. The assembled XM1 tail tube and sheath contain 18 hexameric rings of gp7 and gp6, respectively.

Upon completion of the tail tube and tail sheath assembly, the hexameric tail terminator protein, gp5, attaches to the last rings of gp7 and gp6. The gp5 protein stabilizes the tail tube and sheath structures and creates a binding interface for the independently assembled phage capsid.

The XM1 capsid assembly mechanism is likely similar to that of other tailed dsDNA phages [43]. First, a capsid precursor called the prohead is assembled, which consists of an inner scaffold and an outer shell. The shell of the XM1 prohead consists of 415 copies of the major capsid protein, gp54, and the portal protein dodecamer, gp49, located at one of the vertices. In some phages, such as HK97, the N-terminal region of the major capsid protein acts as the inner prohead scaffold. However, the corresponding region of XM1 MCP is very short (~30 residues), suggesting that XM1 has a separate major scaffolding protein whose gene has yet to be identified. As the MCP is probably cleaved after prohead assembly, the inner prohead scaffold probably also contains several copies of a prohead protease. A possible candidate for this protease is gp46, which has sequence homology to the DNA-dependent metalloprotease Spartan (PDB: 6MDW).

Once the prohead is assembled, the protease digests the major scaffolding protein and cleaves off the N-terminal region of MCP [43,44]. The degradation products escape from the prohead’s interior to liberate a place for genomic DNA. The DNA is then packaged into the prohead through the portal by an ATP-driven packaging machine made of gp48 (as identified by its sequence homology to T4 gp17) [43]. During genome packaging, the gp54 capsid shell expands in volume to accommodate DNA. The expanded capsid shell is decorated by Dec, which reinforces the capsid against pressure created by the packaged DNA. After the completion of genome packaging, the gp48 machine dissociates from the portal vertex. The portal is then sealed by gp1 and gp4 proteins, which prevent DNA leakage. This completes capsid assembly and creates the binding interface for tail binding.

The XM1 tail attaches to the capsid via interactions between the tail terminator protein, gp5, and the connector protein, gp4. Later, the collar spike protein, gp40, attaches to gp4 and gp5 proteins, reinforcing the head and tail interactions (Figure 5f). This completes the assembly of the XM1 virion.

Tentative mechanism of infection. XM1 and its Gram-negative *Vibrio* bacterial hosts coexist in a salt water environment. Gram-negative bacteria have two lipid bilayers: the outer membrane and the inner membrane, with a thin peptidoglycan layer present between these two lipid bilayers. The lipopolysaccharides, which consist of lipid A, core polysaccharides, and O-specific antigen, are inserted into the outer membrane and are initially recognized by phages. Many phages possess TSPs, having β-helix folds that recognize O-specific antigens and sometimes digest them [45]. Phage XM1 acquired two sets of TSPs, one on its neck and the other on its baseplate, which may recognize the O-specific antigens around the host and orient the phage particle on the cell surface for infection.

When XM1 spikes become attached to the host cell, the Brownian motion of the phage particle relative to the cell surface may result in a pull (or push) on the TSPs. The gp18 spikes are attached to the baseplate periphery via gp17 subunits, which are parts of relatively rigid (gp16)_2_–gp17 heterotrimers. The (gp16)_2_–gp17 heterotrimers are, in turn, strongly connected to the contractile sheath initiator protein gp15 which, in turn, is tightly attached to the first ring of the sheath. Thus, the push or pull signal may transfer from gp18 tail spikes to the sheath and trigger sheath contraction. As in other CISs, the signal from gp18 most likely also triggers conformational changes in the baseplate, resulting in an expansion of the baseplate wedge ring and detachment of the baseplate periphery from its central hub. In pyocin R2, the (gp16)_2_–gp17 counterpart, (PA0618)_2_–PA0619, is a smaller version of the baseplate wedge [23]. In the post-contracted state, the PA0618 dimer breaks, and the wedge hexamer ring expands. In phage T4, even though the gp6 dimer is still intact in the post-infection states, the diameter of the wedge ring formed by (gp6)_2_–gp7 is also expanded [22,46]. In phage XM1, the baseplate wedge (gp16)_2_–gp17 may expand as observed in other CISs, which may allow the central hub proteins gp11, gp13, and gp14 to attach to the cell wall. During the sheath contraction, the baseplate periphery probably remains attached to the contractile sheath and the cell surface (via gp18 spikes), whereas the central hub proteins, gp11 and gp13, and the cell-piercing protein, gp14, remain attached to the tail tube. The sheath contraction drives the cell-piercing protein and the tail tube through the bacterial cell wall. Then, genomic DNA is injected through the tail tube into the host’s cytoplasm.

## 3. Methods

Phage sample preparations. *Vibrio rotiferianus* was grown at 28 °C with vigorous shaking in marine broth (Marine Broth 2216, Difco Cat. No. 279110). Phage XM1 stock in SM buffer (Tris 50 mM pH 7.5, NaCl 100 mM, and MgSO_4_ 8 mM) was added to the host when it was in the log phase (OD_600_) of growth, with a multiplicity of infection of 0.1. After 3 h of infection, the culture became turbid and foamy. Lysate was separated from the cell debris by spinning down the culture at 3000× *g* for 20 min. NaCl and PEG8000 were added to the collected supernatant to final concentrations of 1M and 10%, respectively. The mixture was stirred slowly at 4 °C overnight and then spinned down at 10,000× *g* for 10 min at 4 °C. The pellet was resuspended in 10 mL of SM buffer. DNase I and RNase A were added to the resuspension in final concentrations of 20 μg/mL and 1 μg/mL, respectively. The solution was incubated at room temperature for 1 h and then centrifuged at 3000× *g* for 10 min to pellet down any insoluble material. In a chloroform-compatible tube, an equal volume of chloroform was added to the supernatant, then mixed by inversion for 2 min. The mixture was centrifuged at 12,000× *g* for 10 min at 4 °C. This supernatant was then carefully transferred to another tube, avoiding the white precipitate layer located between the chloroform and supernatant. This chloroform treatment and centrifugation were repeated two to three times until there was little or no white precipitate between the chloroform and the supernatant. The supernatant was then loaded onto a CsCl density gradient (1.2 g/mL, 1.35 g/mL, 1.45 g/mL, 1.55 g/mL, and 1.7 g/mL) and centrifuged at an average of 111,000× *g* for 3 h at 4 °C. The band located at 1.45 g/mL was collected with a needle punched through the side of the tube. To remove the CsCl, the buffer of the collected phage band was exchanged with SM buffer by centrifugation filtration using Amicon Ultra-15 Centrifugal Filter Units (Millipore, 100 KDa cutoff) five times.

Cryo-EM data collection and image processing. Aliquots of 3.5 μL of the purified phage sample were added onto glow-discharged Lacey carbon (400 mesh, Ted Pella Inc., Redding, CA, USA) EM grids. The grids were blotted for 5 to 7 s at a relative humidity of 80% and then plunge-frozen in liquid ethane using a Gatan CP3 system. The grids were then loaded into an FEI Titan Krios electron microscope operated at 300 kV and equipped with a Gatan K2 Summit detector (3838 × 3710 physical pixels).

Cryo-EM movies were automatically recorded in super-resolution mode using a nominal magnification of 18K (equivalent to 0.81 Å per pixel at the sample level), a defocus range of 0.5–3.5 µm, and a dose rate of 2.87 e^−^/Å^2^/s (equivalent to a total electron dose of 31.59 e^−^/Å^2^). A total of 3014 movies, each composed of 55 frames, were collected. Each frame had an exposure time of 200 ms. The movies were subjected to whole-frame motion correction using a modified version of the MotionCorr2 [47] program. Subsequently, the movie frames were aligned and summed to obtain individual “micrographs” using the Appion program [48]. The contrast transfer function parameters of each micrograph were estimated using the program CTFFIND3 [49]. These micrographs were then used for three-dimensional cryo-EM reconstructions.

Single-particle three-dimensional reconstructions. From each micrograph, particles were manually boxed according to the targeted region for reconstruction (e.g., the head, neck, baseplate or full tail) (Appendix A). Reconstruction of the icosahedral head was performed using the jspr software package [50]. Five sets of 200 particles were randomly selected and assigned random orientations to build the initial models. Each of the five models was then refined by cycles of particle alignment using projection matching methods. This process was considered successful when at least two of the resultant maps converged to the same structure. This structure was then used as an initial model to calculate the reconstruction using the entire cryo-EM dataset. For the structures of the phage neck and baseplate, *cryoSPARC* software (version 2.12) [51] was used to build initial models and to refine the structures. These structures were further refined using the program *jspr*. To achieve near-atomic resolution, the anisotropic magnification distortion, defocus, astigmatism, scale, and beam tilt for individual particle images were refined [52]. The final resolution of each reconstruction was determined by the point where the Fourier shell correlation between two independent half-sets fell below 0.143 [53].

The 6-fold-symmetric reconstruction of the distal part of the tail, which included the baseplate and a part of the tail tube surrounded by the contractile sheath, reached a 3.2Å resolution. The 6-fold-symmetric reconstruction of the phage portal and neck region reached a resolution of 3.6Å, as did the 12-fold-symmetric reconstruction of the portal. The icosahedral averaged reconstruction of the XM1 capsid resolved at a 3.2Å resolution. The 6-fold-symmetric reconstruction of the contracted tail of the empty XM1 particles had a resolution of 9.1Å. The 6-fold-symmetric reconstruction of the full-length non-contracted tail of XM1 particles had a resolution of 10.6Å.

Model building and structural analysis. The sequence of every individual XM1 open reading frame was submitted to the HHPred protein homology search server [21]. For some XM1 proteins, like the major capsid protein gp53, their homologs with known structures were detected with high confidence, as measured by the probability score (Appendix A). For those proteins, the atomic model of the homolog (Appendix A) was used as a starting model or guideline to trace and build an initial XM1 model using the program Coot [54]. For proteins that did not show high homology to known structures, poly-alanine chains were built in the density maps, and large amino acid side chains were used as landmarks to assign sequences. Later, atomic structures of the amino acid side chains were built to obtain initial models of the proteins. All initial models were improved by multiple rounds of manual rebuilding in Coot and computational real-space refinement using the Phenix software package [55]. Symmetry-related protein subunits were then placed into the density maps using the Phenix.apply_ncs program. Finally, the protein complexes were refined against cryo-EM density using the Phenix real-space refinement tool [56] (Appendix A). Interactions among proteins were analyzed using the program PISA [56] from the CCP4 software suite [57].

The 9Å-resolution reconstructions of the tail of an empty XM1 phage with a contractile sheath was mainly used for analysis of the conformational changes of sheath protein, gp6. The density corresponding to one sheath subunit was extracted using the Chimera [58,59] segmentation tool, and the map was converted to the Situs software format. The extended sheath protein structure, obtained using the 3.2Å-resolution baseplate map, was used as an initial model. This model was fitted into the extracted density by the ddforge program from the Situs package [28], which allows for flexible fitting into medium-resolution maps. The fitting results show that subunits of the sheath protein mainly move as rigid bodies during tail contraction. However, the N- and C-termini of the sheath protein showed significant conformational changes and were fitted into the density using Coot.

## Figures and Tables

**Figure 1 viruses-15-01673-f001:**
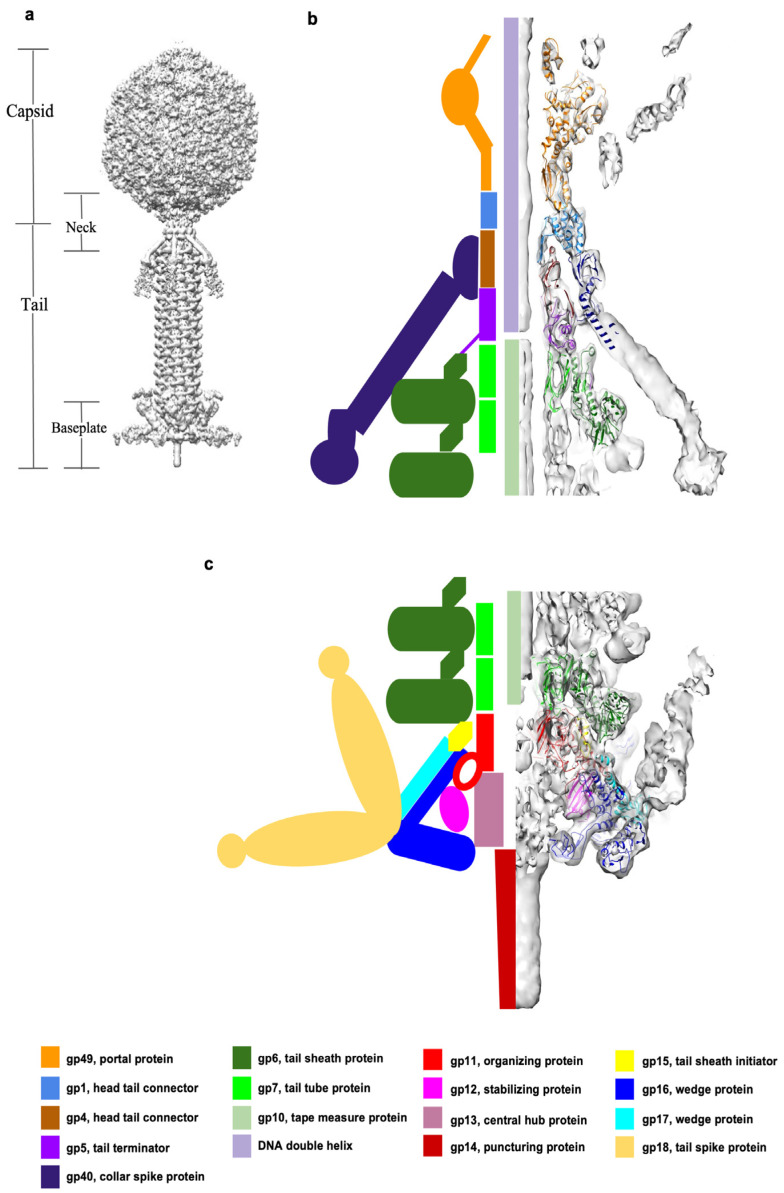
Overview of phage XM1 structure. (**a**) Cryo-EM reconstructions of the icosahedral capsid and the 6-fold-symmetric tail were joined together to generate a model of the entire XM1 virion. (**b**) The neck region of the XM1 virion. The left side shows a schematic diagram representing the neck proteins. The right side shows the backbones of the proteins (color-coded as noted below the panel) in the cryo-EM map of the neck region (grey). (**c**) Structure of the XM1 baseplate. The left side shows a schematic diagram representing the baseplate proteins. The right side shows the backbones of the proteins (color-coded as noted below the panel) in the cryo-EM map of the baseplate region (grey). Protein color codes and their functions are provided at the bottom of the figure.

**Figure 2 viruses-15-01673-f002:**
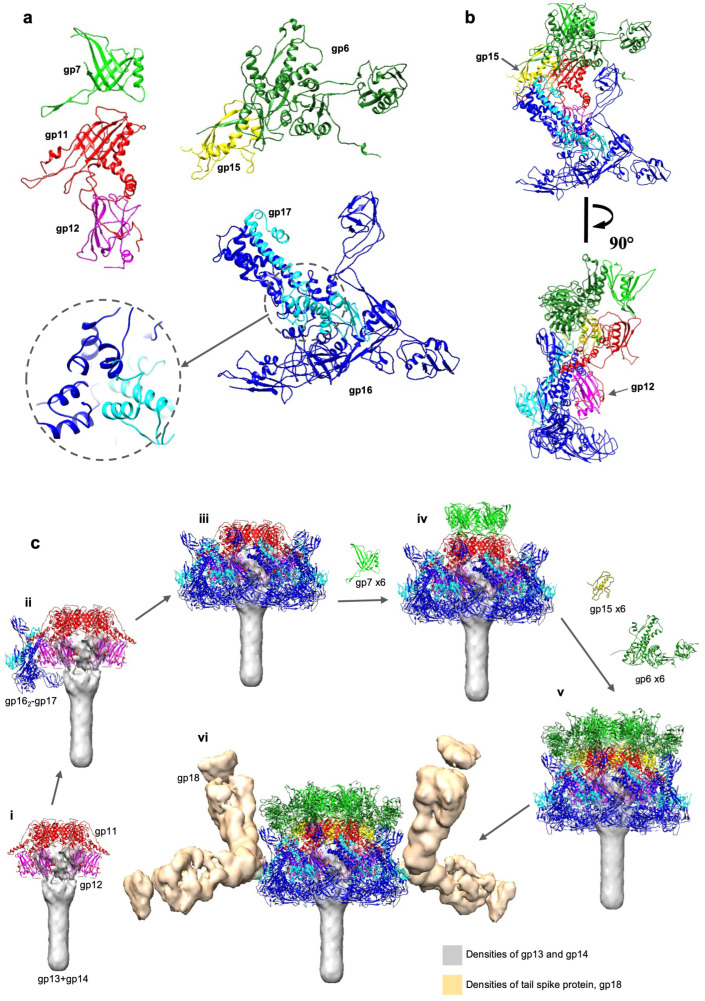
Structures of the tail proteins and the proposed baseplate assembly pathway. (**a**) The ribbon diagrams of gp6 (forest green), gp7 (green), gp11 (red), gp12 (magenta), gp15 (yellow), gp16 (blue), and gp17 (cyan). The trifurcation region of (gp16)_2_–gp17 is outlined with a dashed circle. (**b**) The asymmetric unit of the baseplate is shown in two orientations, rotated by 90° relative to each other. The sheath initiator protein, gp15, indicated by an arrow in the top orientation, attaches to gp11 and interacts with wedge proteins and the sheath protein, gp6. The gp12 baseplate stabilization protein, indicated by an arrow in the lower image, strongly interacts with gp11 and gp16. (**c**) Proposed assembly pathway of the XM1 baseplate. The grey density, segmented from the 6-fold-symmetric map of the full tail, represents the hub protein, gp13, and the cell-puncturing protein, gp14. (**i**) The central baseplate organizing protein, gp11, assembles on top of the hub protein, with gp12 subunits attached to it. (**ii**) The (gp16)_2_–gp17 complex is bound to the central hub by interactions with gp11 and gp12. (**iii**) The six baseplate wedges are then fastened together by dimerization of the gp16 subunits from neighboring wedges. (**iv**) The first hexameric ring of the tail tube protein, gp7, assembles on top of the gp11 ring. (**v**) The sheath initiator protein, gp15, attaches to the baseplate, and the first hexameric ring of sheath protein, gp6, assembles on the top of the baseplate. (**vi**) The density (gold), segmented from the map of the full tail, corresponds to the tail spike protein, gp18, which attaches to the baseplate periphery.

**Figure 3 viruses-15-01673-f003:**
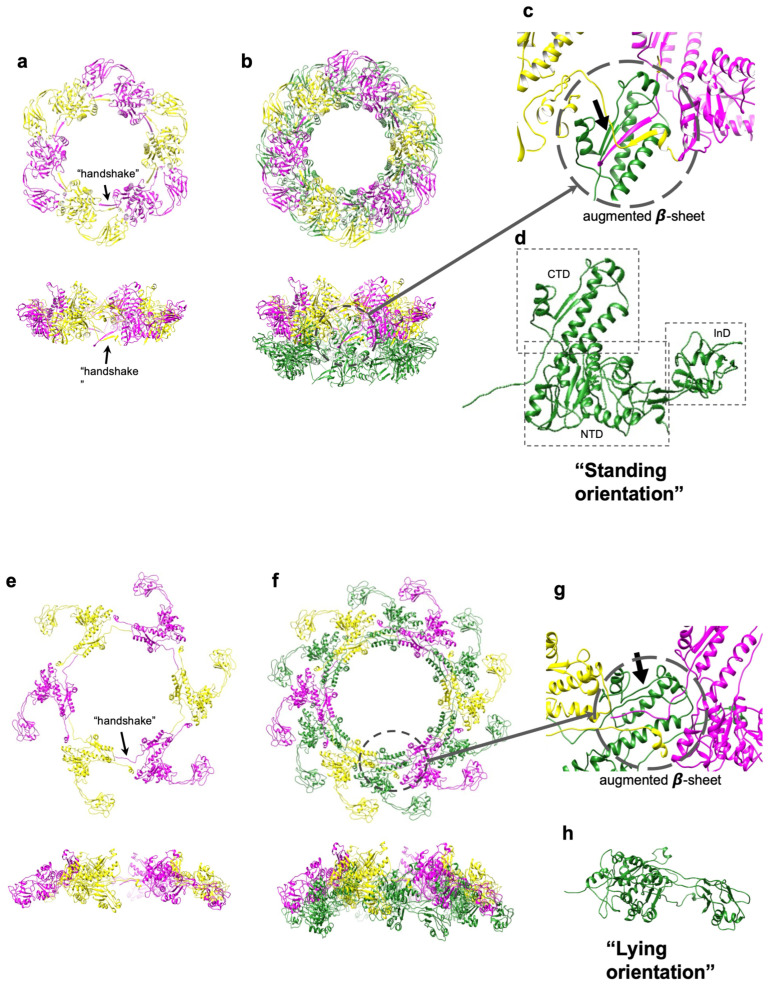
Structure of tail sheath protein, gp6, in the extended and contracted sheaths. (**a**) Top and side views of one hexameric gp6 ring of the extended sheath. The gp6 subunits are colored in magenta and yellow alternately to show the “handshake” between the N- and C-termini of two adjacent subunits. (**b**) Top and side views of two hexameric gp6 rings of the extended sheath. The upper ring is colored as in panel A and the lower ring is shown in green. (**c**) Enlargement showing that the “handshake” β-strands from the upper sheath ring form an augmented β-sheet with β-strands of the C-terminal domain (CTD) from the lower ring. The augmented β-sheet is indicated by an arrow. (**d**) Side view of a single gp6 subunit of the extended sheath. This view represents the “standing” orientation of the sheath subunit. The CTD, NTD, and InD domains of gp6 are outlined by squares. (**e**) One hexameric ring of the sheath in the contracted state. The gp6 subunits are colored as in Panel A to show that the “handshake” between adjacent subunits is conserved after sheath contraction. In the contracted state, the sheath ring is expanded when viewed from the top and compacted when viewed from the side. (**f**) Two hexameric rings of the contracted sheath with the lower ring shown in forest green. (**g**) Enlargement (circled) showing that the β-sheet augmentation is conserved in the contracted sheath. (**h**) Side view of one gp6 subunit of the contracted sheath. This view represents the “lying” orientation of the sheath subunit.

**Figure 4 viruses-15-01673-f004:**
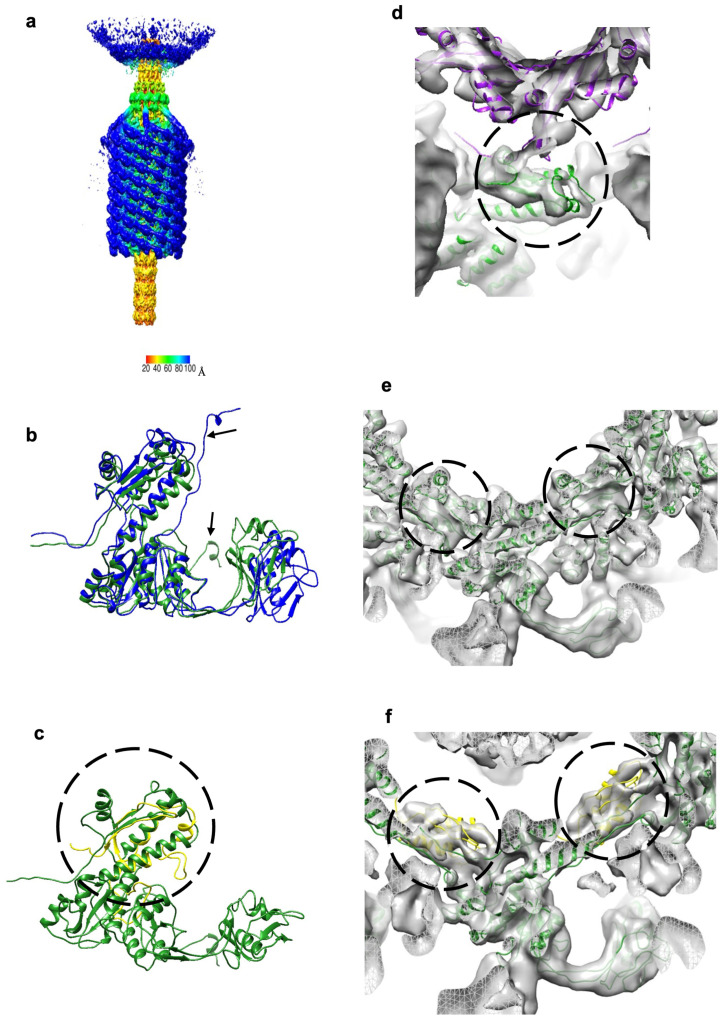
The contracted sheath structure and its interactions with tail terminator protein, gp5, and sheath initiator protein, gp15. (**a**) The 6-fold-symmetric reconstruction of the XM1 tail calculated using phage particles with contracted tail sheaths and empty capsids. The density is colored according to the distance from the central vertical 6-fold axis (see color bar). (**b**) The sheath protein, gp6, in the contracted state (blue) is superimposed onto the sheath protein in the extended state (forest green). Note that the N-terminus shows the most conformational changes as pointed out by arrows. (**c**) The sheath initiator protein, gp15 (yellow and circled), is aligned with the C-terminal domain (CTD) of the sheath protein, gp6 (forest green). (**d**) The tail terminator protein, gp5 (purple), and the sheath subunits, gp6 (forest green), fitted into the map of the contracted tail. The gp5–gp6 interactions are outlined by the circle. After sheath contraction, the gp6 subunits change their positions relative to gp5. (**e**) Sheath protein subunits fitted into the reconstruction of contracted tails. Densities representing the augmented β-sheets are encircled. (**f**) Sheath initiator protein subunits, gp15 (yellow and circled), fitted into the density next to the first ring of the contracted sheath.

**Figure 5 viruses-15-01673-f005:**
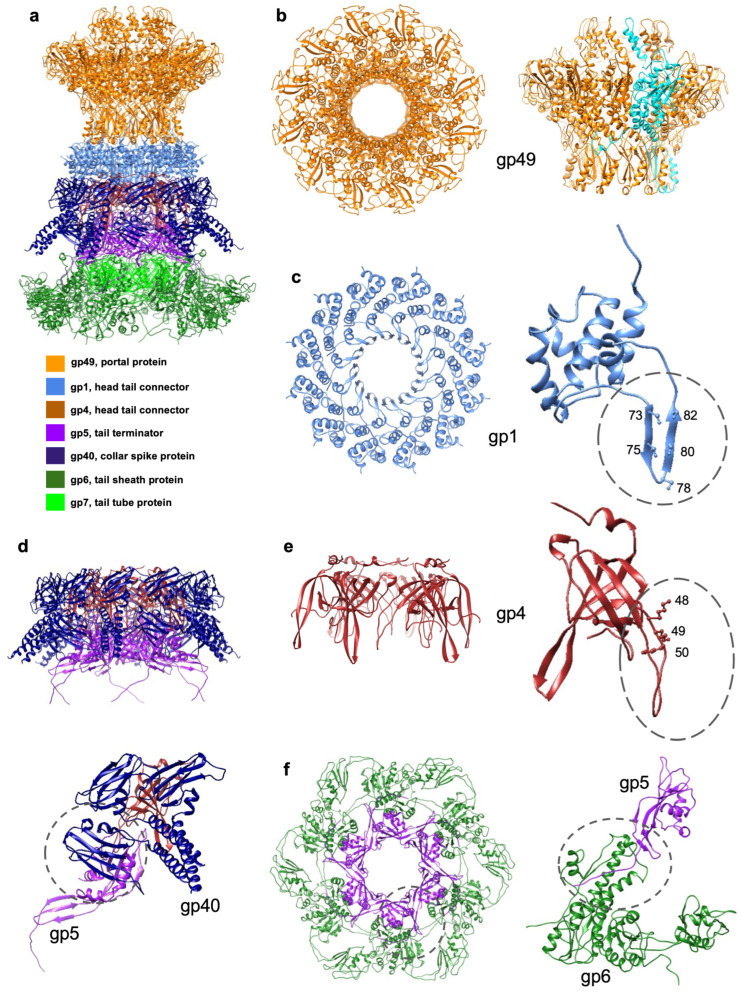
Structure of the XM1 neck. (**a**) The portal protein, gp49, is colored orange. The head completion proteins, gp1 and gp4, are colored light blue and brown, respectively. The tail terminator protein, gp5, is colored purple. The tail sheath protein, gp6, is forest green, and the tail tube protein, gp7, is green. The collar spike protein, gp40, is dark blue. (**b**) The top and side views of the portal protein, gp49. In the side view, one of the chains of the gp49 dodecamer is colored cyan to show that the N-terminal region of the protein wraps around four neighboring subunits. (**c**) Top view of the dodecameric gp1 ring and side view of one gp1 subunit. The β-sheet insertion, facing the DNA helix, is outlined by a circle, and the positions of five serine residues are labeled. (**d**) A complex of the gp4, gp5, and gp40 rings, viewed from the side, is shown at the top of the panel. The gp40 protein reinforces the head–tail interactions. One gp40 trimer, interacting with one gp4 subunit and one gp5 subunit, is shown at the bottom of the panel. (**e**) Side view of the hexameric gp4 ring and of one gp4 subunit. The extended loop between β2 and β3 is circled and the three lysine residues belonging to this loop are labeled. (**f**) A top view of the rings of the tail terminator protein, gp5, and the sheath protein, gp6, is shown on the left. A side view of one gp5 subunit, and one gp6 subunit is shown on the right. The C-terminus of the tail terminator protein, gp5, extends and forms a β-sheet with β-strands of the C-terminal domain of the gp6 sheath protein. The augmented β-sheet is outlined by the circle.

**Figure 6 viruses-15-01673-f006:**
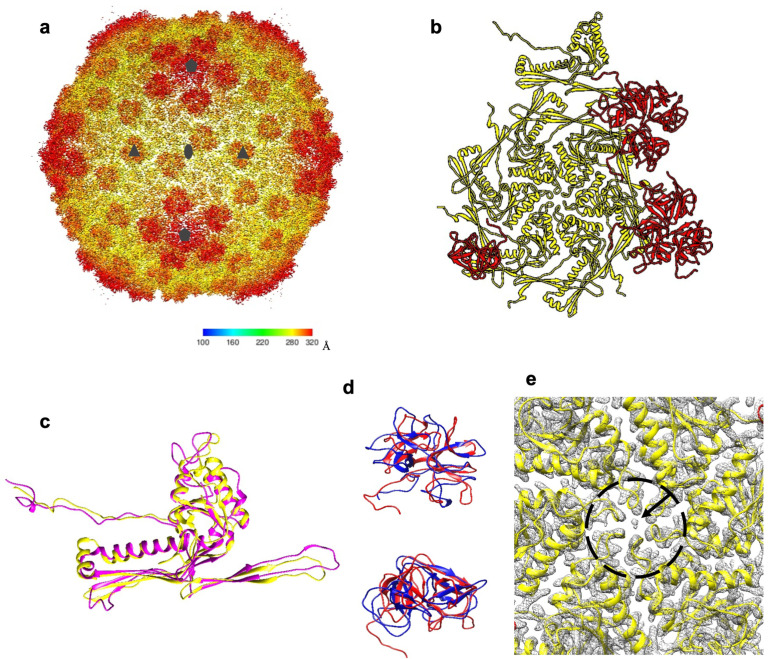
Structure of the XM1 capsid. (**a**) Cryo-EM reconstruction of the icosahedral XM1 capsid. The density is colored according to the distance from the capsid center (see the color bar). (**b**) One asymmetric unit of the capsid. The major capsid protein, gp54, is colored yellow, and the decoration protein, gp53, is colored red. (**c**) The XM1 gp54 (yellow) superimposed onto the HK97 major capsid protein (magenta). (**d**) Side and top views of the XM1 decoration protein (red) superimposed onto the phage TW1 decoration protein (blue). (**e**) Close-up view of the cryo-EM map (grey mesh) near the center of the hexameric gp54 capsomer. The arrow points to the unidentified density in the center of the capsomer.

**Table 1 viruses-15-01673-t001:** Interactions among XM1 proteins ^1^.

		Interface Area (Å^2^)	Free Energy (kcal/mol)
gp1	gp4	1261.7	−18.0
gp1	gp40	322.0	−1.6
gp4	gp5	1395.4	−6.7
	gp40	949.8	−13.0
gp5	gp40	733.4	−2.1
	gp6 (sheath)	1386.5	−16.7
	gp7 (tube)	1157.6	−6.6
gp6 (sheath)	gp6 (same ring)	885.0	−7.6
	gp6 (lower ring)	2897.6	−24.8
	gp7	1138.6	−0.7
gp11	gp6	226.6	−1.2
gp11	gp7	921.5	−9.6
gp11	gp12	1906.3	−15.9
gp11	gp15	1711.1	−8.7
gp11	(gp16)_2_–gp17	1286.1	−13.1
gp12	(gp16)_2_–gp17	1485.4	−10.6
gp15	gp6	1641.9	−16.3
gp15	(gp16)_2_–gp17	1615.4	−15.5

^1^ Data were gathered through PISA analysis [24]. Averaged interface areas and free energy were used. Where there was more than one interface between two different proteins, data in the table reflect the summation of all interfaces.

## Data Availability

The cryo-EM density maps were deposited to the Electron Microscopy Data Bank (EMDB) with the accession numbers EMD-22873, EMD-22896, EMD-22917, EMD-22931, EMD-22959, and EMD-22960. The corresponding atomic models were deposited to the Protein Data Bank with accession codes 7KH1, 7KJK, 7KLN, and 7KMX.

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
