# Peer review of "Structure of *Vibrio* Phage XM1, a Simple Contractile DNA Injection Machine"

_viruses, 2023, doi:10.3390/v15081673_

Round 1

Reviewer 1 Report

This manuscript reports the cryoEM structure of  Vibrio myophage XM1 at 3.2-3.6 Å atomic resolution for the extended form and ~9 Å resolution for the contracted form. This works extends our knowledge of myophages beyond the more “classical” ones as T4 and mu. As this work is well written and perfectly illustrated, I would suggest only minor revisions.

First, I found the structure of the gp1 protein quite interesting, due to the structure of the 24 beta-stranded barrel. Is this barrel structure present in other phages?

Second, the presence of a collar/spike protein is quite interesting. The description of the attachment of gp40 reminds me of the long fibers of siphophage 80alpha described in DOI: 10.1371/journal.ppat.1008314 by Kizziah et al. The authors should compare both structures and cite the paper.

Thirs, do the authors see some density for the TMP? If yes what is its symmetry ( 3 or 6?).

Finally, there is a typo in Figure 5 (gp4; hea instead of head).

Reviewer 2 Report

The authors describe the solution of the structure of Vibrio phage XM1 by cryoEM. The structural work is excellent with few problems or issues. I therefore support its publication. 

However, my main concern is that the description of the possible assembly of the virion (both tail and head) is stated with unbridled confidence, but with little support from the structural work itself. This is starkly reflected in the lack of references in the assembly section (lines 523 - 579). Better justification for the statements is needed. The next section on the mechanism of infection is highly speculative and could be describing any tailed phage with replacing the specific proteins mentioned. This section should be omitted. A better ending for the paper would be comparing this phage to other T6SS systems, highlighting the structural similarities and differences.

Minor comments:

I miss a list of abbreviations.

Line 155:

The manually built proteins have homologs as shown in Supplementary Table 1. Were these used in as guides to build these proteins? At 3.2 Å some detailed features may be difficult to interpret.

Lines 181, 226-243:

The two copies of gp16 appear to differ in conformation. If that is that true, please add more detail? 

Figure 2 & lines 523-537:

The implication in Figure 2c is that it represents the sequence of assembly. Any evidence for this particular scheme?

Line 382:

Same question as for the baseplate proteins (line 155).
